# Nucleic Acid Aptamers in Nanotechnology

**DOI:** 10.3390/biomedicines10051079

**Published:** 2022-05-06

**Authors:** Valentina V. Sinitsyna, Alexandre A. Vetcher

**Affiliations:** 1Nanotechnology Scientific and Educational Center, Institute of Biochemical Technology and Nanotechnology, Peoples’ Friendship University of Russia (RUDN), Miklukho-Maklaya St. 6, Moscow 117198, Russia; 2Shirshov Institute of Oceanology, Russian Academy of Sciences 36, Nahimovskiy Prospect, Moscow 117997, Russia; 3Complementary and Integrative Health Clinic of Dr. Shishonin 5, Yasnogorskaya St., Moscow 117588, Russia

**Keywords:** oligonucleotides, nucleic acids, nanotubes, nanobiosensors, aptamer aptasensor(s)

## Abstract

Nucleic Acid (NA) aptamers are oligonucleotides. They are unique due to their secondary and tertiary structure; namely, the secondary structure defines the tertiary one by means of affinity and specificity. Our review is devoted only to DNA and RNA aptamers, since the majority of achievements in this direction were obtained with their application. NA aptamers can be used as macromolecular devices and consist of short single-stranded molecules, which adopt unique three-dimensional structures due to the interaction of complementary parts of the chain and stacking interactions. The review is devoted to the recent nanotechnological advances in NA aptamers application.

## 1. Introduction

The secondary and tertiary structure of RNA and DNA is defined by their sequence. The structure eventually defines their affinity and specificity [1]. NA aptamers, short (25 to 60 nucleotides) single-stranded DNA or RNA molecules are specially designed to bind to a specific target molecule. Thus, they could serve as components of macromolecular devices for fundamental research and technological applications. Each aptamer adopts unique three-dimensional structures due to the interaction with complementary parts of the chain and stacking.

In 1990, the SELEX method was proposed to indicate the selection of RNA ligands against T4 DNA polymerase. It employed “in vitro selection” by selecting RNA ligands against certain organic dyes [2]. Recently, the call for such ligands or nanodevices components caused the avalanche growth of aptamers application reports. Let us take a look at the progress in this direction in the last decade. Single-walled (swCNT) and multi-walled (mwCNT) carbon nanotubes, which require various aqueous suspensions dispersed with the addition of surfactants or DNA, can be fractionated into components by agarose gel electrophoresis (AGE) [3]. NA aptamers could wrap swCNT. The calculations suggest the stacking involvement. This study initiated the search for the aptamers that could be employed in swCNT manipulations, e.g., to separate metallic swCNTs from semiconducting by AGE [3].

## 2. Results and Discussion

### 2.1. Fractionation of Nanoobjects

In the review, nucleic acids that were treated with ultrasound were considered, with their use, fractionation [4] of a mixture of nanotubes, a mixture of nanotubes on metal and semiconductors in an agarose gel was achieved. Presumably, fractionation can be improved with the use of aptamers with a certain sequence. However, these works were not continued. This stage has not been completed yet.

#### 2.1.1. Treatment Methodologies for Infectious Diseases

Dendrimers [5] are polymers composed of molecules for the treatment and prevention of infectious diseases, but it should be noted that they have proven themselves in medical sciences, nanotechnology, biological technology, and modern industrial technologies. Some potential applications of dendrimer’s are being explored in the early stages of intensive research, including their use in medicine. It has not been determined what the metabolism of these polymers are, how they were taken, perhaps in the form of oral preparations, and what effects the dendrimer metabolites may have on cells, tissues and organs. Although clinical translation is still difficult, improved manufacturing techniques and surface modifications of dendrimers may facilitate clinical translation. 

A dendrimer is a branched polymer, which resembles the branches of a tree (dendron in Greek). A number of surface groups polymer that can be used to couple active ingredients doubles with bifurcation of each terminal end of the polymer or generation (G) of the dendrimer. Typical structures of polyamidoamine (PAMAM) dendrimers [6] with different number of generations are shown in Figure 1. An increase in G from 0 (native PAMAM) to 4 leads to an increase in the number of surface groups from 4 to 64. This branching somewhat increases the average diameter of the PAMAM dendrimer (in most cases, about three times), but significantly increases the average molecular weight of the dendrimer (up to 30 once).

#### 2.1.2. siRNA-Based Diagnostics

siRNA is a 21–25 nucleotide long RNA duplex with two unpaired overhanging nucleotides at the 3′ ends. Each chain has a phosphate group at the 5′ end and a hydroxyl group at the 3′ end.

There is a new perspective of delivering siRNA into wrapped DNA/RNA-containing nanoparticles [7] called “DNA baskets”. In binary DNA/RNA nanoparticles, DNA will serve as a biodegradable “basket”, which is needed to enhance its interaction with the miRNA duplex of a certain sequence. The “DNA basket” will contribute to the impossibility of miRNA degradation in the bloodstream and will serve as the basis for the placement of covalently linked ligands (i.e., targeting protein fragments or antibodies) that direct the DNA basket with its miRNA load to certain types of human body, cells. DNA-miRNA nanoparticles can be composed as dsRNA-DNA triplexes. This allows the use of additional covalently bound components that modify the structural features of the DNA basket and promote the biocompatibility of the therapeutic nanoparticle.

#### 2.1.3. Non-Viral Nanoscale Delivery of Antisense Oligonucleotides 

Such therapy does not allow the development of resistance of cancer cells and increases the effectiveness of chemotherapy, which cannot be achieved with the separate use of individual components. This represents the basis for a new type of cancer therapy based on the simultaneous delivery of an anti-cancer drug and suppressor of hypoxia-inducible factor 1α (HIF1A) [8].

The good impact of cancer chemotherapy is limited by the development of a large tumor drug resistance. An important mechanism of cancer resistance to chemotherapy is the adaptation of cancer cells to tumor hypoxia. Hypoxia occurs in large tumors from the early stages of tumor development as a result of insufficient oxygen supply due to exponential cell proliferation and blood supply. As a result, glucose deprivation and oxygen deficiency play a bimodal role in continued tumor development and survival. In large cases, hypoxia is an unfavorable prognostic sign of cancer, as it is associated with tumor progression and resistance to therapy.

Modified asymmetric PCR method allows synthesis of long ssDNAs composed of tandem repeats of the repetitive vertebrate telomeric sequence (TTAGGG)_n_, and is also applicable to arbitrary (repetitive or nonrepetitive) DNA [9].

### 2.2. Different Types of Nanobiosensors

#### 2.2.1. Aptamer Folding

Some types of RNA enzymes (ribozymes) have been found in biological systems and bred in the laboratory. The variety of RNA enzymes and the similarities between DNA and RNA mean that DNA can also function as an enzyme. However, this DNA enzyme has not been found in nature. The decision was made to adopt the metal-dependent DNA enzyme using the in vitro selection methodology shown in Figure 2.

Aptamers [10] have become promising types of biorecognition for the development of biosensors. Use of Dissipation Controlled Quartz Crystal Microbalances (MKKM-D) for enantioselective detection of a Target Molecule with a Low Molecular Weight (less than 200 Da) by sensors based on special aptamers. QCM-D is a powerful method for label-free characterization and real-time quantification of molecular interactions, and detection of small molecules in contact with immobilized receptors remains a challenge. The conformational change of the aptamer during binding causes the displacement of water acoustically bound to the sensitive layer. Therefore, this phenomenon leads to a significant enhancement of the detection signal.

Impedimetric biosensors are currently being developed [11]. Impedimetric methods are performed for methods of biosensor preparation and prediction of catalyzed enzymatic reactions or measures for the biomolecular determination of specific binding proteins, lectins, receptors, nucleic acids, whole cells, antibodies or antibody-bound drugs. In this interesting direction, little time is devoted to this direction of the research.

#### 2.2.2. DNA Aptamers for Therapy and Diagnostics

Since the first monoclonal antibody was obtained in the 1970s, antibodies have been successfully and widely used in the diagnosis and therapy of human diseases. Aptamers are expected to achieve the same success as antibodies. Due to their stability, low cost and easy manipulation, DNA aptamers will continue to be actively studied and applied. RNA aptamers possess increased stability. RNA aptamers have the superiority in providing heterogeneous 3D structures, which is useful for selecting aptamers with high affinity for difficult targets needed for disease therapy. DNA aptamers may have more promising applications in in vivo diagnostics, while RNA aptamers may have promising applications in therapy. It is believed that in medicine, therapy and diagnostics based on DNA aptamers have great potential for wide application. Before they can be widely applied, therefore, there are still many problems that require further analysis in the future [12]. 

#### 2.2.3. Impedimetric Biosensors

The flexible array lactate biosensor in Figure 3 is made on the immobilization of l-lactate dehydrogenase (LDH) and nicotinamide adenine dinucleotide (NAD+) on a nickel oxide (NiO) film, the average sensitivity of which is increased by using graphene oxide (GO) and a magnetic field (MB beads). With GO and MB, it shows very good sensitivity (45.397 mV/mM) with a linearity of 0.992 from 0.2 mM to 3 mM. According to the results of electrochemical impedance spectroscopy (EIS), the electron transfer resistance of the LDH-NAD+-MBs/GPTS/GO/NiO film was much lower than that of the LDH-NAD+/GPTS/GO/NiO and LDH-NAD+/GPTS films/NiO, and it showed good electron transfer ability. Research is ongoing on detection limits, anti-interference effects and bending tests.

The introduction of bioreceptors in biosensors contains a limiting limit, problem immunization with small antigens and low chemical and thermal resistance. This means that there is a desire to replace bioreceptors’ artificial sensors. Molecular imprinted polymers (MMPs) are stable and reliable, in fact, which makes them easy to use at extreme pressures, temperatures, pH or organic solvents. They are still cheap to manufacture and can be kept dry. MMPs are made not only for small organic molecules: pesticides, amino acids, steroids and sugars, but also for proteins and cells.

A new type of biosensor, in which the configurations of the dielectric properties of the upper part of the electrode are recorded, was launched in the late 1980s. The binding of the analyzed drug with the immobilized affinity component can be found precisely by the label or indicator reaction. Configurations can be attributed to the method of measuring electronic capacitance or impedance, with support for interdigital electrodes or, more often, with support for potentiostatic methods.

Capacitive biosensors belong to the group of affinity biosensors, which are performed by registering direct binding between the surface of the sensor and the desired molecule. This image of the biosensors determines the types of dielectric properties and/or the thickness of the dielectric layer at the electrolyte/electrode interface [13]. Capacitive biosensors have been successfully used to detect proteins, nucleotides, heavy metals, saccharides, small organic molecules and microbial cells. At present, a method of microcontact imprinting has been developed to create sufficiently sensitive and selective biorecognizing cavities on the plane of capacitive electrodes.

Some time ago, developments in biochemistry and molecular biology led to a better understanding of the role of peptide-based probes and showed that their role is much wider than originally thought. This led to the new horizons of biosensing applications, where oligonucleotide-based biosensing could be potentially be replaced by peptide-based approaches which can have an unparalleled impact on molecular diagnostics. The increasing demand for enhanced efficiency and to overcome some of the drawbacks of using oligonucleotides or antibodies has enabled biochemists to come up with synthetic analogues of antibodies called Affimers (Figure 4), which have led to increased projection of biosensing approaches.

Aptamers have become a promising component of biological knowledge in the development of biosensors [14]. A conformational change in the aptamer on directional binding, which causes the displacement of water, is acoustically tied to the sensitive layer. This means that such an appearance leads to a large delay in the detection signal.

#### 2.2.4. G-quadruplex Forming Aptamer

The discovery relates to the field of nanobiotechnology and molecular medicine and concerns DNA aptamers to the thrombin exosite I interacting with prothrombin. These compounds can be used to create medicines that stop intravascular thrombus formation. Here are aptamer-based biosensors for detecting lateral flow [15]. Aptamers are very useful and have biorecognizing substances for the development of biosensors. Aptamers are small, single-stranded pieces of DNA used to select protein targets. A selection method (SELEX) was developed, which allows the isolation of motivated nucleic acid molecules from a large set (more than 1015) of individual molecules, called a combinatorial library [16,17]. Some aptamers in Figure 5 have every chance of not only recognizing their own targets, but also to destroy their biological vigor.

The ability of an aptamer to recognize a target molecule is based on its spatial structure, which allows the aptamer of *E. coli*, such as a protein antibody, to identify the epitope on the plane of the protein. Now, aptamers are considered as promising pharmacological products. Aptamers are not inferior to monoclonal antibodies in the specificity and affinity of interaction with target molecules [18]. As pharmacological products, they have a number of significant advantages over antibodies.

The chemical aptamer product [19,20] (E-AB) for real-time insulin monitoring is a sensor in which a redox labeled G-rich aptamer takes place, which is formed in the G-quadruplex for unique insulin detection. To develop a reproducible E-AB detector using insulin aptamer probes for insulin detection requires pretreatment with 10% sodium dodecyl sulfate (SDS), which is critical because it damages interstitial G-quartets. Subsequently preparatory processing of the detector with 10% SDS achieves a smoother resonance of the detector. When an insulin target is introduced, steric hindrances caused by binding quantitatively decrease the efficiency of electron transfer to the distal redox markers, which actually leads to a rapid change in the signal in the direction of ~60 s. The test demonstrates that insulin E-AB exhibits a detection limit of 20 nM and can be used to discriminate both glucagon, for example, and somatostatin in Krebs-Ringer bicarbonate buffer typically used in perfusion assays. These results demonstrate, in fact, that this test contains the potential for a rapid, unique and quantitative analysis of insulin.

#### 2.2.5. High-Performance Screening of SELEX RNA Aptamer with FACS Using Liposomes

The SELEX method is a method to identify and select the aptamers. It occurs by exponential enrichment in the systematic evolution of ligands. The library of oligonucleotides is gradually enriched with sequences with increased affinity for the target molecule. In classical SELEX [21], the process of creating target incubation, elution and amplification of the binding sequence is repeated until the vast majority of the stored pool consists of target binding sequences. Screening of the RNA aptamer by SELEX with FACS takes place using liposomes. For FACS-SELEX [22], one or more fluorescently labeled antibodies with specificities for desired subpopulations can be used for the enriched separation of the aptamer-bound fraction by FACS, allowing for increased specificity of selection. Since aptamers bind non-specifically to dead cells, cell-SELEX is susceptible to triggering non-specific bindings of aptamers to dead cells, and skewing or biasing the selection process. However, the combination of SELEX pre-enrichment by aptamer screening using FACS [23] allows you to select aptamers with the ability to light up.

Purified SELEX based on aptamer targets includes metal ions, small molecules, proteins and more. In the general flow of SELEX determinations of target aptamers, this is consistent with the original procedure, where distribution with target oligonucleotides is an important step. Therefore, various solid phase carriers such as magnetic nanoparticles (MNPs), agarose beads, nitrocellulose membrane and dosing microplates are used in purified selexes on targets.

#### 2.2.6. Ultrasensitive and Highly Specific Recognition Aptasensors with Different Detection

Such aptasensors set the precedent that single-stranded nucleic acids quickly and firmly bind to graphene sheets using non-covalent π-π stacking and hydrophobic interactions between aromatic nitrogen-containing causes in aptamers and sp-bonded carbon atoms in graphene. In this mode of non-covalent binding, aptamers have every chance to stay on the plane of the graphene sheet, as a result of which any dye conjugated to the aptamer is located in a specific proximity to graphene. The fluorescent dye label is quenched by graphene supported by a nonradiative dipole–dipole bond. In associated targets, aptamers bind with the highest affinity and fold into specific structures. The aptamer-target complex [24] would disrupt the interaction between the aptamer and graphene, resulting in their release from the graphene sheet. This will increase the distance between the donor (dye) and quencher (graphene), leading to the restoration of the fluorescent signal. Switching fluorescent signals from the off state before the target changes to the on position after detecting a motivated aptamer provides a “signal” to the aptasensors for analyte detection. In good FRET [25,26,27] graphene aptasensors, fluorescence recovery is based on a 1:1 binding strategy since one target molecule triggers the release of one labeled aptamer from graphene. The detected fluorescence will become proportional to the concentration of motivated molecules. Subpopulations of RNA [28] molecules for more detailed information that specifically bind to various organic dyes are selected from a population of randomly sequenced RNA molecules [29].

Aptamers showed some original properties, in the amount of long shelf life, simple transformation to ensure covalent bonds with material surfaces, minor batch configurations, profitability and low susceptibility to denaturation. These features have led to the necessary efforts to develop detectors based on aptamers in Figure 6, popular as aptasensors [30], which are designated as optical, electronic and mass-sensitive depending on the signal transmission mode. 

Sensitive and accurate determination of cardiac troponin I (cTnI) is of final importance for the diagnosis of myocardial infarction. A chemical (EC) biosensor [31] based on a double aptamer has been developed for cTnI detection based on DNA nanotetrahedra (NTH) capture probes and functional hybrid nanoprobes. NTH-based Tro4 aptamer probes were attached to the plane of a screen-printed gold electrode (SPGE) via Au-S association, making improved spatial magnitude and accessibility for cTnI capture. Next, hybrid nanoprobes were prepared using Fe_3_O_4_ magnetic microparticles as nanoparticles to load large amounts of cTnI-specific Tro6 aptamer, natural horseradish peroxidase (HRP), HRP-imaging Au and Pt nanozymes and G-quadruplex/hemin DNAzymes. These signaling nanoprobes recognize motivated cTnI based on the Tro6 aptamer and amplify signals to increase detection sensitivity while supporting enzymatic processes. The excellent enhanced effect of the EC signal can be explained by the cumulative catalysis of hybrid nanoenzymes, HRP and DNAzyme. The cTnI target was sandwiched between two types of aptamers (Tro4 and Tro6) at the electrode interface [32,33].

#### 2.2.7. DNA Origami as a Nanosensor

Folding of DNA molecules DNA origami is needed in a nanosensor to analyze the enzymatic energy of hAGT DNA repair. The method uses conformational configurations that cause the interaction of α-thrombin with DNA APTAMERS, and illustrates the introduction of DNA origami in Figure 7 as a biosensor for protein repair [34]. 

The programmability and qualities of DNA self-assembly provides ways of neatly organizing matter at the nanoscale. DNA origami [35] connects two-dimensional and three-dimensional DNA structures and is needed to organize biomolecules, bionanophotonic and electrical components with a resolution of 6 nm. Two-dimensional DNA origami is needed as a platform for the organization of other chemical compounds, which, as a result, could be located on technologically important substrates. Perhaps these layouts were used only with DNA nanostructure to hold chemical particles on a plane, and as is known, they have never been used to immobilize nucleic acid structures on surfaces with a resolution of less than 10 nm, providing a platform for these probable applications, such as multiplex biochemical tests, to create metasurfaces with potentially reconfigurable functions.

#### 2.2.8. G-quadruplex Thrombine-Binding DNA Aptamers

31-mer aptamer of DNA oligonucleotide RA36 more effectively inhibits the coagulant activity of thrombin compared to the well-known aptamer 15TGT (thrombin-binding aptamer). The RA36 aptamer has a two-pattern structure that includes two G-rich regions capable of forming a G-quadruplex. The circular dichroism method shows that the RA36 aptamer forms an antiparallel G-quadruplex similar to the 15TGT G-quadruplex. The thermal stability of the G-quadruplex RA36 is significantly lower than that of 15TGT under physiological conditions (the concentration of the stabilizing cation is 5 mM). The double quadruplex structure of RA36 is confirmed by the CD spectra of the deletion mutants, i.e., the G-quadruplex can be formed by both the first and the second G-rich site of the RA36 aptamer [36].

#### 2.2.9. G-quadruplex Antithrombin Aptamers

In the presented objects, we used the simplest 15-mer G-quadruplex antithrombin aptamers [37,38] HD1 and its analogs, namely RA36 as a covalent dimer of HD1 and GL2-HD1 as a non-covalent dimer with a G-quadruplex lock. HD1 and RA36 are single-stranded G-quadruplexes, while GL2-HD1 forms non-covalent (GL2-HD1)2 dimers in a concentration dependent manner, giving ~50% dimers at 1 mmol dm^−3^.

The oligomerization of aptamer HD1 affects its pharmacokinetics, and therefore, requires a thorough consideration during the development of new aptamer-based antithrombotic drugs. In general, a simple and robust technique is necessary for the quantification of various oligomeric and conformational states of G-quadruplexes. The currently used approach is the size exclusion high performance liquid chromatography (SE HPLC); however, its common protocols lack a linear correlation between the chromatographic retention volume and the logarithm of molecular weight for monomeric G-quadruplexes, the reason being probably the sub-optimal separation conditions. As a result of such complications, the molecular mass of aptamer HD1 was twice overestimated. Analytical ultracentrifugation is used as an additional method for the estimation of G-quadruplexes oligomeric [39].

#### 2.2.10. NA Aptamers with GFP Chromophore for Biosensors 

In cell and molecular biology, the GFP gene is often used as an expression reporter. It has been used in modified forms to create biosensors, and many animals have been created that express GFP, demonstrating that the gene can be expressed throughout a given organism, in particular organs or cells of interest. GFP can be introduced into animals or other species by transgenic methods and stored in their genome and the genome of their offspring. To date, GFP is expressed in many species, including bacteria, yeast, fungi, fish and mammals, including human cells.

Biosensors based on GFP fusion proteins are powerful tools for real-time observation of events in living cells. Insertion of GFP into another protein has made it possible to obtain biosensors capable of signaling intracellular events through intrinsic fluorescence changes, resonant fluorescence energy transfer and changes in subcellular localization. The difficult task of finding the correct insert site to obtain a biosensor can be accelerated by screening libraries of random GFP inserts [40].

#### 2.2.11. DNA Aptamers Binding to M11

The in vitro method directly uses live bacterial cells and the strategy of systematic ligand evolution by exponential enrichment (SELEX). Individual aptamer sequences, if screened based on their binding to 10 M-types, can be used as targets. Aptamer pools obtained during 5–8 rounds of SELEX showed high affinity for *S. pyogenes* target cells. Several aptamer sequences preferentially bind to the M11 M-type of *S. pyogenes*. The improved SELEX method with bacterial cells has been successfully used to generate aptamers that are selective for S. pyogenes and some of its M-types. These aptamers are potentially useful for detecting *S. pyogenes*, deriving binding profiles of various M-types, and developing new M-typing technologies for non-specialist laboratories or point-of-care testing [41].

#### 2.2.12. Impedance Detection Methods for Miniature Analytical Systems

Sensor systems based on impedance detection can be divided into two types depending on the presence or absence of a biorecognition element in them. Systems of the first type, which are called impedimetric biosensors (ID-biosensors), fix the change in impedance caused by the binding of the target and bioligand (antibody, antimicrobial peptide or aptamer) on the surface of the electrodes. Detectors of the second type are called impedimetric sensors (ID-sensors), and register the change in electrical impedance as a result of chemical and metabolic processes occurring in the sample volume and directly on the surface of the electrodes. System impedance analysis detection as efficient and effective method in miniature analytical systems has a number of developments in the field of sensors based on impedance detection and the possibility of integrating them in laboratories on a chip (LNC) [42].

#### 2.2.13. Multianalytical Analysis

Deciphering the binding mechanisms between aptamers and small ligands is critical for improving and optimizing existing applications and developing new ones. Of particular interest is the mechanism of enantiospecific binding of small molecules to unstructured aptamers. One archetypal example is the chiral linkage between 1-tyrosinamide and its 49-mer aptamer (A49), for which there is no structural or mechanistic information. The advantage of the strategy is multiple analytical characterization (i.e., using electroanalytical methods).

The results allow us to conclude that the enantiospecific binding reaction occurs by the mechanism of induced fit, when the ligand promotes the stage of primary binding of nucleation near the 5′-end of the aptamer, followed by directed folding of the aptamer around its end, targeting from the 5′-end to the 3′-end. Functionalization of the 5′-end by a chemical label, poly(dA) tail, protein or surface affects the kinetic/thermodynamic constants up to two orders of magnitude in the limiting case of a surface-immobilized aptamer [43]. Lists of aptamers with the scope of their application are listed in Table 1.

## 3. Conclusions

Previously, layers of DNA oligonucleotides were used for aptamers of nanobiosensors, but now, more modern ones have appeared. Current chemical biosensors target two structural proteins: spike and nucleocapsid with antibodies-based biorecognition molecules, while the E and M proteins are considered unusual biomarkers of early viral infection. Biorecognition molecules include polyclonal/monoclonal antibodies, aptamers, single-stranded RNAs or DNA oligonucleotides. Electrochemical biosensors detect lightning-fast responses and can be integrated with on-board labs to discover superior analytical platforms at the point of service [44]. This scheme is the most common detection procedure without any bioconjugation of label lines. It is necessary to do cyclic voltammetry, differential pulse voltammetry, rectangular voltammetry, etc., if the biorecognizing molecule (antibody, aptamer, small organic molecule, etc.) is conjugated with a small electroactive synthesis, for example ferrocene derivatives.

Aptamers using dendrimers (siRNA method) are used for the treatment of infectious diseases and against cancer. DNA and RNA aptamers have their application in the treatment of diseases (molecular diagnostics of diseases in vivo)—they are able to signal intracellular events inside the human body. A dual aptamer based on DNA-nano tetrahedron (NTH) capture probes and functional hybrid nanoprobes is used to accurately determine cardiac troponin I (cTnI) for the diagnosis of myocardial infarction.

Biosensors must detect viral infection in all types of clinical specimens and are subject to rigorous testing. This direction in the study of nanobiosensors is called single molecular detection. The use of nanobiosensors in medicine and biology opens up new possibilities for the prevention and treatment of many well-known and widespread diseases.

## Figures and Tables

**Figure 1 biomedicines-10-01079-f001:**
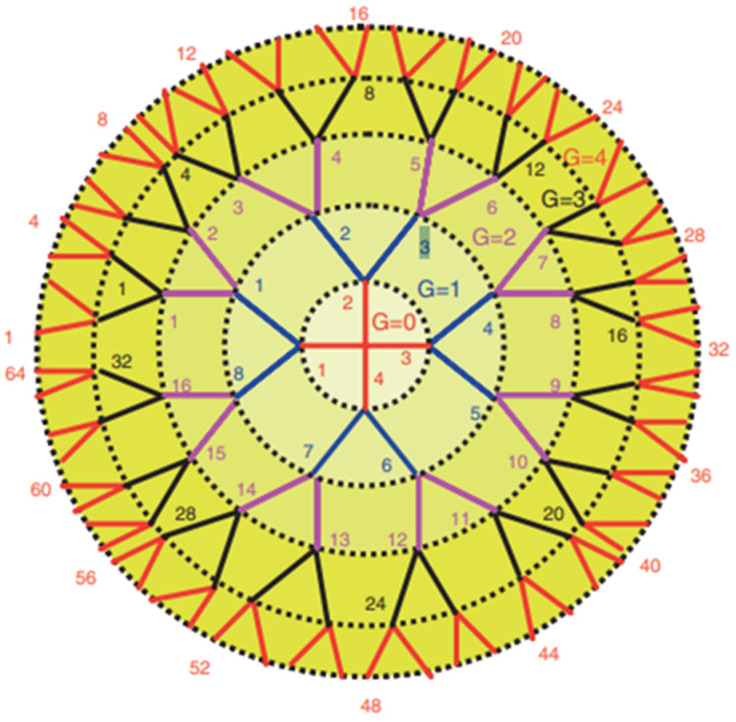
A schematic of PAMAM dendrimer. G0, G1, G2, G3, G4—generations of dendrimer.

**Figure 2 biomedicines-10-01079-f002:**
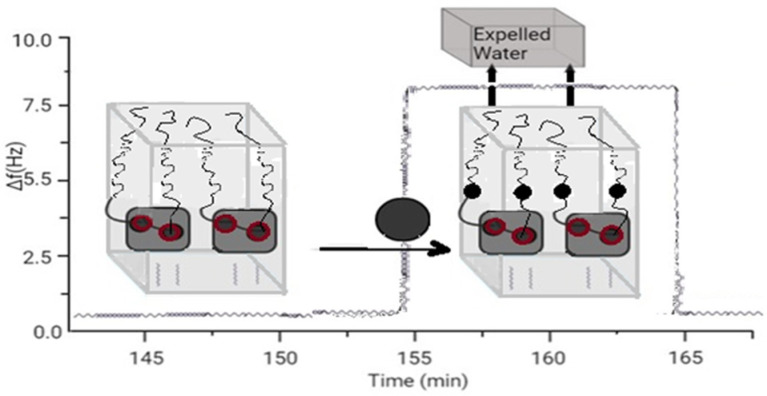
Sensor based on aptamer’s folding.

**Figure 3 biomedicines-10-01079-f003:**
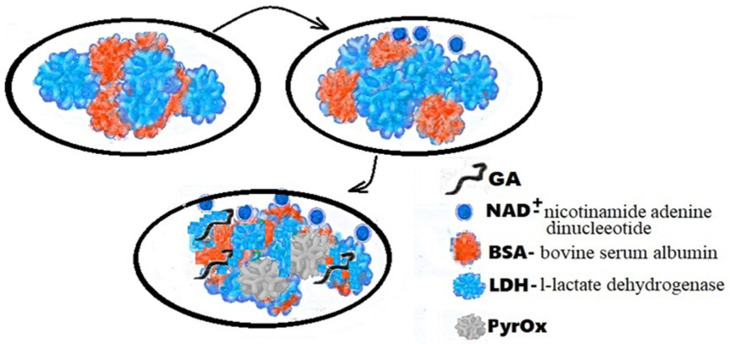
Schematic representation of LDH-NAD+/PyrOx biosensor GA-BSA, LDH.

**Figure 4 biomedicines-10-01079-f004:**
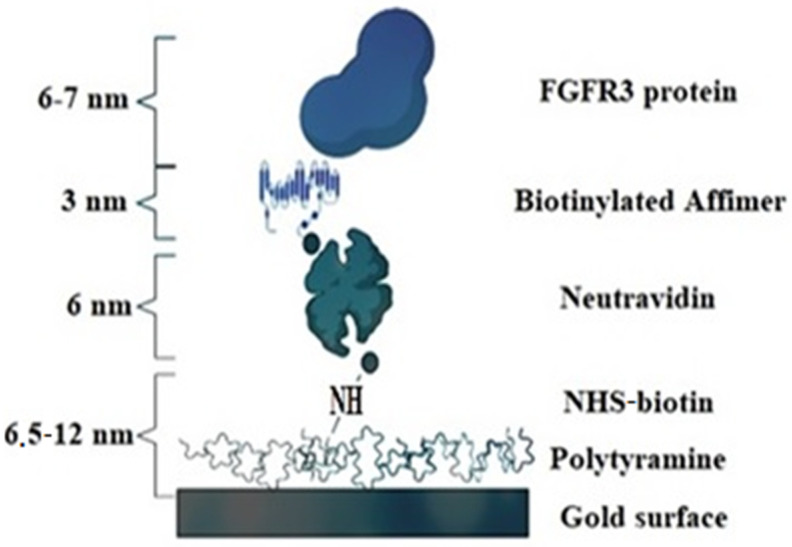
Schematic representation of a fully-fabricated Affimer-based biosensor.

**Figure 5 biomedicines-10-01079-f005:**
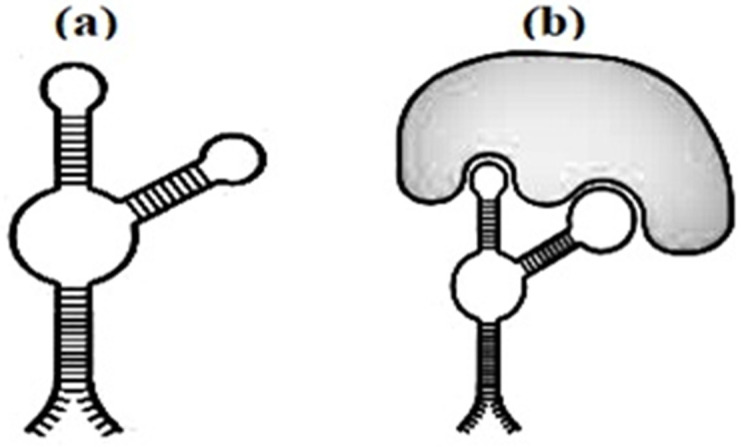
The secondary structure of the aptamer (**a**) and aptamer with marker (**b**).

**Figure 6 biomedicines-10-01079-f006:**
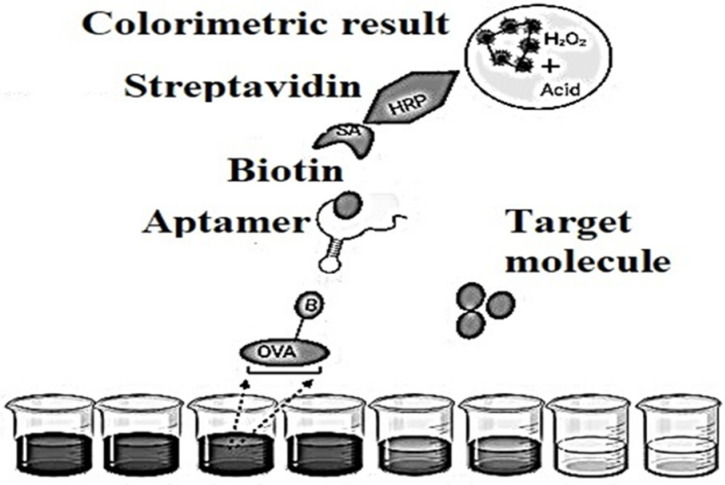
Schematic representation of an indirect competitive enzyme-linked aptamer assay (ELAA)-based aptasensor.

**Figure 7 biomedicines-10-01079-f007:**
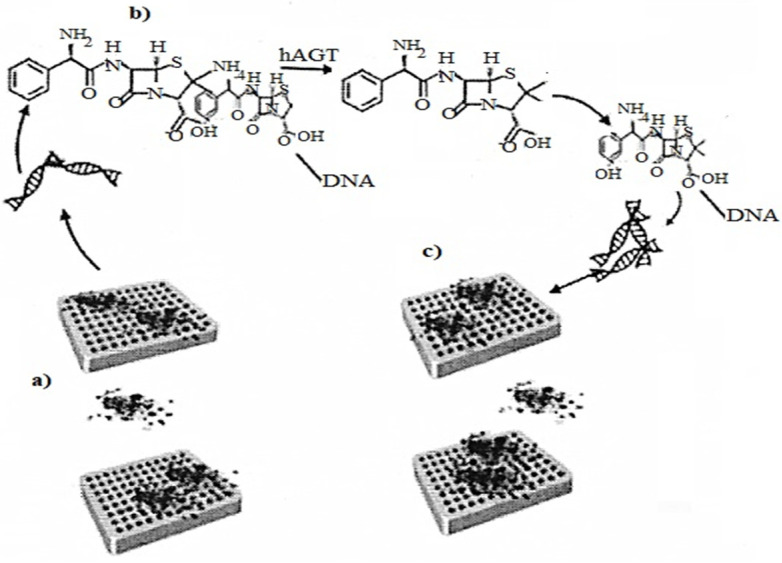
(**a**) Representation of the asymmetric binding of a-thrombin to TBA aptamers of methylated DNA origami; (**b**) Methyl-TBA repair by hAGT, thus G-quadruplex formation; (**c**) Representation of the symmetric binding of a α-thrombin to the repaired DNA origami quadruplexes.

**Table 1 biomedicines-10-01079-t001:** List of aptamers with application areas.

	Aptamer-Based Method	Aptamers	Application Areas
1	siRNA [7,17]	Dendrimers	Treatment of infectious and non-infectious diseases.
2	SELEX [41]	DNA aptamers binding to M11	Diagnostics.
3	FACS-SELEX [22,23]	RNA-aptamers	Cancer-affected cells detection.
4	DNA origami [34,35]	DNA-aptamers with α-thrombin	Multiplex biochemical tests.
5	Liposomal drug delivery system [8]	HIF1A	Improving the effectiveness of chemotherapy.
6	QCM-D-method [43]	Apt_49_	DNA-aptamer-based bio-recognizable sensors. Therapy. Visualization of proteins.
7	Circular dichroism method [36]	RA36	Therapy with thrombin inhibitors.
8	HPLC size exclusion [37,38,39]	HD1, GL2-HD1	Prevention of thrombosis.
9	EC biosensor [40,42]	DNA nanotetrahedron aptamer (NTH)	Analysis of cTnI for the diagnostics of heart attack.
10	Transgenic methods [40]	NA aptamers based on the GFP	Detection of intracellular events throug fluorescence changes.

## Data Availability

Not applicable.

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
