# Peer review of "Nucleic Acid Aptamers in Nanotechnology"

_biomedicines, 2022, doi:10.3390/biomedicines10051079_

Round 1

Reviewer 1 Report

Aptamers have high affinity toward metal ions, small molecules, peptides,  proteins, tissues and cells. They have several advantages over antibodies such as ease of evolving, high stability, ease of storage etc. The review article is timely and highlights recent advances. I have a few points to make. 

1) A brief paragraph about how the aptamers are evolved (capture SELEX, conventional SELEX, FACS based SELEX etc.) should be included.

2) A table containing the list of aptamers against where they used will be highly helpful. 

3) A plethora of RNA and DNA aptamers based on GFP chromophore have been developed and used for biosensors. They should be discussed. 

Author Response

Response to Reviewer 1

Dear Reviewer:

First of all, let us thank you for your efforts to fit our submission in the best shape for MDPI readers. We 100% agree with your view on points to fix.

We consider all of your comments seriously and, please, find below the list of our efforts:

1) A brief paragraph about how the aptamers are evolved (capture SELEX, conventional SELEX, FACS based SELEX etc.) should be included.

The SELEX-explaining Section “2.3.5. High-performance screening of SELEX RNA aptamer with FACS using liposomes” is added

2) A table containing the list of aptamers against where they used will be highly helpful. 

The “Table 1. List of aptamers with application areas” is added

3) A plethora of RNA and DNA aptamers based on GFP chromophore have been developed and used for biosensors. They should be discussed. 

The Section “2.3.10. NA aptamers with GFP chromophore for biosensors” is added

We hope that we were able to address all of the listed issues.

Regards

Dr. Alex Vetcher

Reviewer 2 Report

The topic of the manuscript sounds interesting but, overall, is not addressed well, in my opinion. Aptamer is the keyword, but the manuscript starts with many sections about nanotubes and nanoparticles that have interactions with some kinds of aptamers, although not very well described.

The manuscript put together many pieces of information but they are not well-organized. Actually, on the whole, it is hard to see a clear connection among the different sections.

English is another problem. I saw several grammar mistakes and typos. Moreover, some sentences are not clear at all.

The "Conclusions" section appears detached from the rest of the manuscript.

I think the author should first clarify what they want to present in this review paper, find a proper title and organize the different sections in a logical way. Moreover, the manuscript shall be checked by a native English speaker.

Author Response

2022-04-28

Response to Reviewer 2

Dear Reviewer:

First of all, let us thank you for your efforts to make our submission attractive to MDPI readers. We 100% agree with your view on points to fix.

We consider all of your comments seriously and, please, find below the list of our efforts: 

  •       The topic of the manuscript sounds interesting but, overall, is not addressed well, in my opinion.  – We carefully rewrite the body and hope that in the present shape it looks much better
  •       Aptamer is the keyword, but the manuscript starts with many sections about nanotubes and nanoparticles that have interactions with some kinds of aptamers, although not very well described.– We omit the Section, devoted to carbon nanotubes, leaving only a brief notation in the Introduction. 
  •       The manuscript put together many pieces of information but they are not well-organized. Actually, on the whole, it is hard to see a clear connection among the different sections. – We assume that after the exclusion of a couple of sections and re-writing the submission looks more concise.
  •       English is another problem. I saw several grammar mistakes and typos. Moreover, some sentences are not clear at all. – After re-writing, we invited a native American speaker to clarify all unclear spots.
  •       The "Conclusions" section appears detached from the rest of the manuscript. – It was re-written
  •       I think the author should first clarify what they want to present in this review paper, find a proper title and logically organize the different sections. – We change the title to “Advances in nucleic acid aptamers application”
  •       Moreover, the manuscript shall be checked by a native English speaker. - We invited native American speaker Mss Vasilisa Bystrykh to make it as clear as possible.

Regards

Dr. Alex Vetcher

Round 2

Reviewer 2 Report

The manuscript has been improved considerably with respect to the initial submission.

Just a few minor comments:

  • line 11: I would define explicitly the meaning of NA;
  • line 15: remove "etc.". It is present also in other sentences in the main text;
  • line 22 "these" should be "this".

Author Response

2022-05-04

Response to Reviewer 2

Dear Reviewer:

First of all, let us thank you for your careful analysis of the latest version of our submission.

We tried to correct the body following your comments:

  • line 11: I would define explicitly the meaning of NA;

We added the definition and also explanation, that our submission is devoted to DNA and RNA aptamers

  • line 15: remove "etc.". It is present also in other sentences in the main text;

Done in lines 15 and 27.

  • line 22 "these" should be "this".

Done.

We hope that now the submission gained the best possible shape

Regards

Dr. Alex Vetcher
